# Innovative Strategies in Microvascular Head and Neck Reconstruction

**DOI:** 10.3390/medicina59071194

**Published:** 2023-06-24

**Authors:** Z-Hye Lee, Tarek Ismail, John W. Shuck, Edward I. Chang

**Affiliations:** 1Department of Plastic Surgery, University of Texas MD Anderson Cancer Center, Houston, TX 77030, USA; zlee@mdanderson.org (Z.-H.L.); jwshuck@mdanderson.org (J.W.S.); 2Department of Plastic, Reconstructive, Aesthetic and Hand Surgery, University Hospital of Basel, 4031 Basel, Switzerland; tarek.ismail@usb.ch

**Keywords:** innovation head and neck reconstruction, alternate donor site, virtual surgical planning, immediate dental implants

## Abstract

The field of reconstructive microsurgery has witnessed considerable advancements over the years, driven by improvements in technology, imaging, surgical instruments, increased understanding of perforator anatomy, and experience with microsurgery. However, within the subset of microvascular head and neck reconstruction, novel strategies are needed to improve and optimize both patient aesthetics and post-operative function. Given the disfiguring defects that are encountered following trauma or oncologic resections, the reconstructive microsurgeon must always aim to innovate new approaches, reject historic premises, and challenge established paradigms to further achieve improvement in both aesthetic and functional outcomes. The authors aim to provide an up-to-date review of innovations in head and neck reconstruction for oncologic defects.

## 1. Introduction

The field of reconstructive microsurgery was founded on principles described by Alexes Carrell, which laid the foundation for the field of vascular surgery and, eventually, transplant surgery [1,2,3]. Over time, these foundations ushered in an era of microvascular surgery, allowing the reconstruction of complex defects that previously could not be adequately corrected, resulting in disfiguring and suboptimal outcomes [4]. The use of free flaps has not only emerged as the gold standard for the reconstruction of a variety of different defects but has also proven to be superior to pedicled flaps [5,6,7,8]. While these locoregional options are important to incorporate into the algorithmic approach to head and neck reconstruction, microsurgery has revolutionized our approach from simply trying to achieve a closed wound to truly maximizing patient satisfaction and restoring both form and function. A detailed and thorough comparison of microvascular free flaps to local, regional pedicle flaps is beyond the scope of the current review, which aims to focus on some of the innovations in microvascular head and neck reconstruction.

Numerous studies have demonstrated high success rates with microvascular reconstruction of head and neck defects [9]. As such, the reconstructive microsurgeon can now focus on the nuances of refining the reconstruction and innovating new strategies to improve outcomes. In fact, aside from simply considering the closure of the defect, standards for achieving the most optimal aesthetic outcomes are becoming increasingly important as well, where patients’ quality of life and satisfaction are affected by which donor site provides the aesthetic and functional result [10]. These strategies often incorporate both new and conventional flap options, and the reconstructive microsurgeon must be well-acquainted with both soft tissue and bony reconstruction. Alternate flap donor sites such as the lateral arm or the medial sural artery perforator (MSAP) flap are also gaining more traction and may provide a superior color match and improved donor site morbidity, and should be included in the reconstructive algorithm, particularly in the setting of a prior flap loss, recurrence, or the need for multiple free flaps [11,12,13]. In combination with soft tissue reconstruction, the reconstruction of bony defects has also witnessed tremendous advances, both in terms of technological advancements as well as the incorporation of different donor sites [14]. The use of virtual surgical planning (VSP) and medical modeling has rapidly grown in popularity and is becoming the gold standard in mandible and midface reconstruction at many institutions [15,16,17,18]. In addition, facial paralysis is a challenging problem that continues to require new strategies for improving outcomes, particularly in the oncologic setting. Despite high success rates with facial reanimation using a functional gracilis, patients undergoing a radical resection with a sacrifice of the facial nerve are in desperate need of innovative solutions to address facial paralysis [19]. Finally, the use of artificial intelligence in predicting complications in microvascular head and neck reconstruction is also becoming increasingly important in risk stratification and aims to prevent disastrous outcomes in this complex patient population [20,21]. The present review aims to provide an overview of the innovations that have been described in the literature, many of which have been pioneered at the authors’ institution.

## 2. Soft Tissue Reconstruction

In the field of head and neck reconstruction, a variety of different soft tissue defects can be encountered that range from external skin coverage to volume replacement to the restoration of speech and swallowing function. As such, the reconstructive microsurgeon needs to have a reliable algorithm to address soft tissue defects of the scalp to a parotidectomy and also be able to reconstruct an intraoral or glossectomy defect as well as an esophageal defect [22,23,24,25]. There is little debate regarding the workhorse flaps in head and neck reconstruction, namely the anterolateral thigh (ALT), radial forearm, and latissimus dorsi muscle or myocutaneous flaps have long been the most popular donor sites. Given the reliable anatomy, large caliber vessels, adequate pedicle length, ability to tailor flap size, and minimal donor site morbidity, most defects can be reconstructed with one of these flaps. However, in the field of head and neck reconstruction, patients can develop recurrent disease or complications following treatment, such as osteoradionecrosis, that often mandates another free tissue transfer. In the era of perforator flaps with increased comfort in microsurgery and advancements in technology, it is imperative to have a broader armamentarium of soft tissue flap donor sites that can be used reliably in this complex patient population [26].

In cases when workhorse flaps are unavailable, the reconstructive microsurgeon must have exposure and comfort with alternate donor sites. While used predominantly for breast reconstruction, the profunda artery perforator (PAP) flap is an extremely reliable donor site with reliable anatomy with sufficient pedicle length, although the artery tends to be smaller than the ALT. The perforator anatomy has been well-mapped and anecdotally is more reliable than the ALT perforators [27,28]. Aside from the reliable perforator anatomy, the PAP allows for simultaneous harvest concurrently with the resection and can be harvested in the supine frog-leg position. For head and neck reconstruction, the PAP is best harvested in a longitudinal fashion allowing identification and selection of the largest perforator. When a thinner flap is needed, and the radial forearm is not an option, an ulnar forearm or ulnar artery perforator (UAP) can be considered. The ulnar artery is a large caliber artery comparable to the radial artery; however, careful attention must be paid to avoiding injury to the ulnar nerve. The microsurgeon should also be aware that the pedicle to the UAP is considerably shorter than the radial forearm flap [29,30]. Two additional flaps that can provide thin tissue with a much longer pedicle, similar to the radial artery, while avoiding a skin graft to the donor site are the medial sural artery perforator (MSAP) flap and the lateral forearm flap. It is important to note that the arterial diameter for both of these flaps is typically less than two millimeters, much smaller than either the radial or ulnar arteries [26].

In contrast to the radial and ulnar forearm flap, the donor site of the lateral forearm flap needs to be closed primarily and should not be skin grafted because the lateral epicondyle is often exposed after harvesting the flap. Harvest of the lateral forearm often provides thinner tissue and provides a significantly longer pedicle length comparable to the radial forearm flap. However, the width of the flap is limited to the pinch of the skin at the donor site, which is typically approximately six centimeters [31]. In addition, a lateral arm perforator flap can be harvested, which can provide a bulkier flap by harvesting tissue from the more proximal arm if more volume is needed with intermediate thickness between the ALT and a forearm-based flap, but this does significantly diminish pedicle length [32]. Similar to the UAP flap, the harvest of the lateral forearm or lateral arm perforator flap requires careful attention to avoid injury to the radial nerve during the pedicle dissection.

Regarding the MSAP flap, the perforator location has poor reliability in the authors’ experience. While there are landmarks that serve as a guide for designing the flap, the perforator anatomy is much less reliable than many of the other flaps. Therefore, the MSAP flap is often harvested in a freestyle fashion. Aside from the location of the perforators, the perforators may not arise from the medial sural artery resulting in a much smaller caliber artery, typically less than two millimeters. Aside from the availability of a myriad of donor sites, new technologies such as the use of intraoperative perfusion imaging using indocyanine green have also become routine practice, particularly when more volume is needed to avoid risks of partial flap loss [33,34].

## 3. Virtual Surgical Planning and CAD-CAM

The use of medical modeling and virtual surgical planning (VSP) has become mainstream and is generally now considered the new gold standard in bony head and neck reconstruction at many centers. The earliest descriptions using high-resolution CT scans to plan the resection and reconstruction using a free fibula flap demonstrated proof of concept and paved the way for further advancements and developments using this technology [35,36,37]. The reconstructive surgeon is now able to couple CT angiograms of the donor site with the three-dimensional bony models allowing for detailed planning of both the skin paddle and bony components and ultimately resulting in a more reliable reconstruction [38,39].

With improvements in technology and imaging, VSP and computer-assisted design and computer-assisted modeling (CAD-CAM) have expanded beyond simply providing cutting guides and templates. Patient-specific customized titanium plates can now be milled or printed to intricate and precise configurations without compromising the strength and durability of the construct [40]. Printed reconstruction plates offer the advantage of lower profile design while minimizing the hardware burden (Figure 1). While early experience using VSP may have been prohibitively expensive, some institutions have developed in-house systems rather than outsourcing the design and modeling to commercial vendors [41]. Regardless of the actual costs associated with CAD-CAM, numerous studies have demonstrated the benefits of cost-utility with shorter operative times and potentially improved outcomes [42,43]. However, the technology is not without important limitations that must be considered by the reconstructive surgeon. Aside from the costs associated with the customized models, guides, and plates, there is always the potential for changes that may occur during the resection that may require intraoperative modifications or even render the plans entirely unusable. Recent studies have demonstrated potential long-term complications such as non-union and hardware exposure with customized plates [44,45,46]. Further, the use of CT angiograms for delineating the vascular anatomy also requires appropriate planning and coordination and should not belittle the need for additional imaging and the potential risks of additional radiation and contrast-induced nephropathy.

## 4. Dental Rehabilitation

CAD-CAM technology has also ushered in an era of immediate dental rehabilitation where pre-determined screw holes can be designed to avoid interference with dental implants that are placed at the time of the fibula flap reconstruction. For the majority of patients who undergo a free fibula flap, radiation is often critical for the control of microscopic disease, which increases the risks and complications, often precluding patients from dental implants altogether. While some reports have demonstrated successful implant engraftment into a radiated fibula, other studies have demonstrated increased rates of implant failure in radiated flaps [47,48]. For these reasons, efforts are now aimed at placing dental implants before radiation, allowing patients the potential to achieve complete dental rehabilitation and improving patients’ quality of life [49,50,51]. Recent studies have demonstrated increased success rates with the use of VSP in dental implant placement and integration compared to traditional approaches [52].

Another noted benefit of CAD-CAM mandible reconstruction was fortuitously noted in the setting of a posterior mandibulectomy where the condyle has been sacrificed. Historically, reconstruction was commonly performed using a soft tissue flap simply to restore the volume; however, patients, unfortunately, suffered from a greater degree of malocclusion as well as a suboptimal cosmetic result [53,54]. With the use of CAD-CAM technology, precise placement of the fibula into the glenoid fossa has been able to improve dental occlusion without increased risks of trismus and has now changed the paradigm for the reconstruction of posterior mandibulectomy defects at the authors’ institution [55,56]. In addition, the use of a medial femoral trochlear (MFT) flap has been performed in combination with a fibula osteocutaneous flap to reconstruct the mandibular condyle and may further improve outcomes [57] (Figure 2). The MFT flap is harvested to provide vascularized cartilage that is anastomosed to the distal end of the fibular flap pedicle as a flow-through flap. Providing vascularized cartilage in combination with the fibula osteocutaneous flap creates an entirely vascularized construct and better restores normal occlusion, and allows patients to progress to normal oral intake.

## 5. Nerve Reconstruction and Reinnervation

Aside from the advancements made in bony reconstruction, strategies have been investigated to address other deficits resulting from a composite mandibulectomy. The inferior alveolar nerve and its distal continuation, the mental nerve, are routinely sacrificed with the resection, which results in sensory deficits in the chin and lower lip. Sensory restoration of the mental nerve may now be achieved through nerve coaptation with the use of an intervening nerve allo- or autograft (Figure 3). While studies have demonstrated spontaneous recovery of lower lip sensation following a mandible resection [58], advancements in nerve repair and restoring sensation have proven effective in other aspects of head and neck reconstruction, such as in the reconstruction of glossectomy defects where the creation of a sensate flap has proven superiority in optimizing patients’ speech and swallowing function. A sensory nerve can routinely be harvested with the flap, such as the lateral antebrachial cutaneous nerve with the radial forearm flap or the lateral femoral cutaneous nerve with the ALT flap. Despite the deleterious effects of post-operative radiation therapy, subtle improvements in sensory restoration may provide potential benefits in improving patients’ quality of life. Early outcomes with mental nerve reconstruction have demonstrated promising results at the authors’ institution [59].

## 6. Alternate Bone Flaps

The use of VSP has also revolutionized the reconstruction of the midface. Custom-3D printed titanium plates are also used for multi-segment stabilization of bone flap reconstruction of the maxilla and reconstruction of the orbit (Figure 4). Just as restoration of proper alignment and occlusion is critical in optimizing functional outcomes for the mandible, the establishment of precise positioning of the globe is vital to restoring proper binocular vision for patients who undergo an orbitomaxillectomy. The advent of customized 3D-printed titanium plates based on patients’ preoperative imaging maximizes the chances of achieving normal vision [60]. However, hardware burdens in the midface and peri-orbita are at increased risk for delayed hardware infection given the exposure to and involvement of the nasal and maxillary cavities. The oncologic patient is often at elevated risk for hardware complications, including infection, extrusion, and non-union, given the need for post-operative radiotherapy.

Therefore, vascularized tissue remains the gold standard to minimize issues with hardware infection, exposure, and the need for removal. While the use of non-vascularized bone grafts, such as the rib and iliac crest, is also a reasonable option, our preference is to use vascularized bone flaps in the midface when possible. The fibula and scapula continue to be widely used in osseous flaps in the midface region. A recent addition to the armamentarium is the medial femoral condyle (MFC) flap [61]. The MFC is most commonly used in the reconstruction of extremity defects, in particular, scaphoid non-unions or avascular necrosis of the carpal bones, but its relatively reliable anatomy can also be harnessed for the reconstruction of bony midface defects. The natural contour and curvature of the flap is well-suited for the reconstruction of the orbital floor, but it can also be used for the maxilla, palate, and even for nasal reconstruction. The greatest limitation, however, is the pedicle length which frequently requires the use of vein grafts; however, a common approach that we have utilized is to engineer a chimeric flap using a flow-through orientation with a conventional soft tissue flap such as the ALT flap.

For most midface defects, a soft tissue flap is often needed to fill dead space or provide coverage of the skull base or closure of the palate. In these circumstances, an MFC flap can be connected to a branch of the pedicle of the ALT flap or another fasciocutaneous flap. The distal continuation of the lateral descending circumflex vessels is well-suited in terms of size match and length to serve as the recipient vessels for the MFC. The engineered chimeric flap then allows independent positioning of the MFC flap for the bony reconstruction, while the soft tissue flap can be used to fill the dead space of the maxillary sinus or to reconstruct the palate as needed. The added pedicle length of the ALT flap is generally sufficient to reach the recipient vessels, either the facial vessels at the level of the mandible or the superficial temporal vessels. Early experience using the MFC and the chimeric design has demonstrated high success rates without an increase in donor site morbidity, hospital stay, or recovery time [62].

## 7. Facial Reanimation

The sequelae of facial paralysis from skull base or parotid tumors create significant morbidity, and for patients who undergo a radical parotidectomy, the aesthetic deformity and functional impairments are dramatic. Historically, free flaps were used to correct the volume deficit following a radical resection, while reconstruction of facial nerve defects was achieved using static approaches and nerve grafting [63,64]. Given the relatively older age of most patients as well as the requisite need for adjuvant radiation, serious doubts pervaded the benefit and functionality of a functional muscle transfer in this patient population. Consequently, the primary modality for reconstruction in this setting focused on more conservative approaches. Previous studies have demonstrated restoration of resting tone for some patients with nerve grafts, even in the setting of adjuvant radiation [64]. A static sling can be used as an adjunct to improve resting symmetry and immediate oral competence as well as speech intelligibility by suspending the oral commissure, nasal ala, and nasolabial fold.

Recently, the current paradigm has evolved at our institution to incorporate the use of dynamic reconstruction, including nerve and functional muscle transfers for patients requiring a radical parotidectomy. The novel approach challenges the established premise of forgoing a functional muscle transfer in elderly patients or in patients destined to undergo adjuvant radiation [64,65]. While these factors were considered relative contraindications to dynamic reconstruction in the past, we have confirmed that patients undergoing immediate functional gracilis transfer were able to achieve volitional control. When using a single donor site from the medial thigh, a profunda artery perforator (PAP) flap can be harvested with the gracilis muscle allowing the reconstructive microsurgeon to address the volume deficit using the PAP flap and restore dynamic facial movement using the gracilis muscle [66]. A simultaneous harvest of the PAP and gracilis can be performed in the supine position. When using a longitudinal incision, the PAP and gracilis can be harvested from a single donor site. Under certain circumstances, the gracilis and PAP perforator can even arise from the same pedicle allowing the harvest of a chimeric PAP-gracilis flap which obviates the need for a double free flap and two sets of microvascular anastomoses. While a CTA can be performed to evaluate whether the PAP and gracilis pedicles converge, this has not been the practice at the authors’ institution.

The decision of which nerve to use to power the gracilis muscle remains an area of active investigation. If branches of the facial nerve are available and the proximal stump is clear of a tumor, our recommendation is to use a combined approach with facial nerve grafting and a functional gracilis transfer. Under these circumstances, our preferred recipient nerve is the nerve to the masseter [67]. If the masseteric nerve is not available due to the extent of the resection, other potential recipient nerves are the spinal accessory or the hypoglossal; however, these are only used if they were also sacrificed with the resection. In rare circumstances, an end-to-side nerve coaptation can be performed. While a cross-facial nerve graft can also be performed, this typically would require a staged fashion subjecting a patient to a second lengthy operation, hospital stay, and recovery time for a secondary functional muscle transfer. Our current approach introduces a paradigm shift in performing an immediate functional transfer to achieve dynamic facial reanimation in a single operation using a single donor site.

## 8. Future Directions

Microvascular head and neck reconstruction has witnessed tremendous advancements and innovations over the years, which have been instrumental in improving outcomes for patients. Historical goals have largely become obsolete, and the objective of simply avoiding thromboses and total flap losses has largely been replaced with restoring function, minimizing donor-site morbidity, and maximizing patient satisfaction. For most high-volume institutions, microvascular free flaps have become the gold standard for the reconstruction of extensive head and neck defects and have largely supplanted pedicled flaps. With the growing field of artificial intelligence and augmented reality, patients can have their surgeries planned beforehand to allow for more efficient and reliable reconstruction [67,68,69]. It is important to note that pedicled flaps are still a vital component of the armamentarium of any reconstructive surgeon, especially as salvage options for patients with orocutaneous fistula, hardware exposure, or in the setting of a vessel-depleted neck.

Overall, the field of microsurgery has experienced incredible advancements based both on modern technology, as well as the simple application of technique and anatomic principles set forth by Alexes Carrell and many others over the past century. Innovation in microsurgery continues to progress at a lightning pace, with numerous recent advancements from head to toe. The field of composite tissue allotransplantation has pushed the field of transplant medicine, allowing patients to undergo face and extremity transplantation [70,71]. An under-appreciated and under-recognized problem that plagues many patients undergoing oncologic resection with neck dissection and adjuvant radiation is the dilemma of lymphedema. While conservative measures using facial compression and massage techniques have proven effective, the field of lymphedema super microsurgery can also be applied to head and neck patients [72,73]. The use of a lymphovenous bypass/anastomosis (LVA) or lymph node transfer has been well-documented in the treatment of extremity lymphedema, but its use in head and neck lymphedema is much more limited [74]. Unfortunately, only a limited number of case reports exist using the LVA approach, but early reports are promising [75,76].

Overall, microvascular head and neck reconstruction has undergone tremendous advancement and evolution over the years with improved outcomes and high success rates; however, innovative strategies are still necessary to optimize both the function and aesthetic outcomes for patients undergoing aggressive resections and adjuvant therapies. For soft tissue reconstruction, a myriad of different donor sites are available, but concerns with donor site morbidity are becoming increasingly scrutinized. The radial forearm flap, which was historically the primary option for a thin pliable donor site, has been criticized more frequently in the modern era with the popularity of alternate donor sites that can avoid a skin graft and potential cold intolerance, and impaired function of the upper extremity [77,78,79]. The search for the ideal donor site remains elusive, but likely there is no single donor site that is ideal, and the reconstructive microsurgeon must decide which donor site to use based on the extent and type of defect as well as the patient’s body habitus and available donor sites [80]. Similar considerations exist with bony reconstruction; however, the use of the fibula donor site in combination with CAD-CAM technology has largely supplanted other donor sites for mandible reconstruction [81]. Despite the reported advantages, judicious use and consideration of the limitations and costs of VSP are still important for the operating surgeon, who should critically evaluate and aim to improve outcomes even further [82].

## 9. Conclusions

A number of novel advancements have been achieved in microvascular head and neck reconstruction with the goal of optimizing and maximizing both form and function. Ultimately, it is the responsibility of all surgeons and physicians to continue to innovate and develop novel strategies in order to improve and enhance the quality of life of patients, minimize complications, and optimize outcomes.

## Figures and Tables

**Figure 1 medicina-59-01194-f001:**
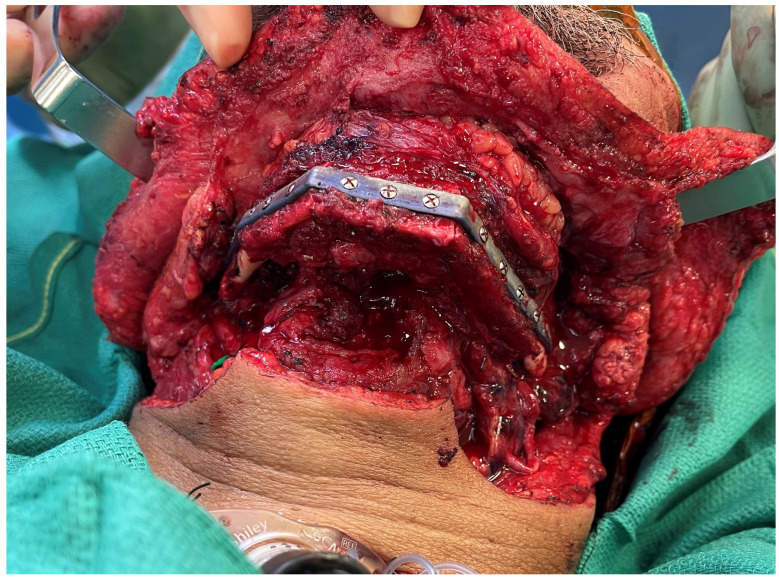
Printed plate with three-segment fibula planned to utilize VSP demonstrating precise osteotomy alignment between segments.

**Figure 2 medicina-59-01194-f002:**
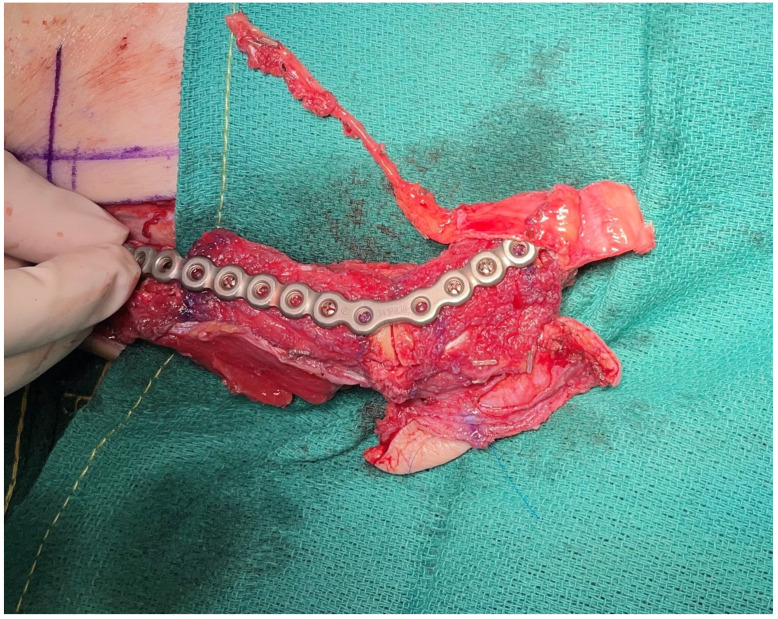
Free medial femoral trochlea flap and free fibula flow through flap for ramus and condylar reconstruction.

**Figure 3 medicina-59-01194-f003:**
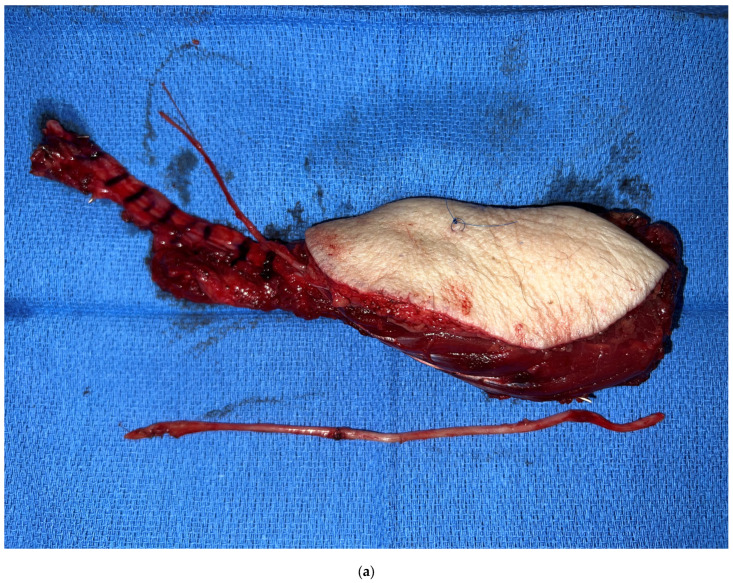
(**a**) Osteocutaneous, sensated fibula flap and sural nerve graft for inferior alveolar nerve reconstruction. (**b**) The inferior alveolar nerve was reconstructed with a sural nerve graft. Distal nerve coaptation is covered with a fibrin wrap.

**Figure 4 medicina-59-01194-f004:**
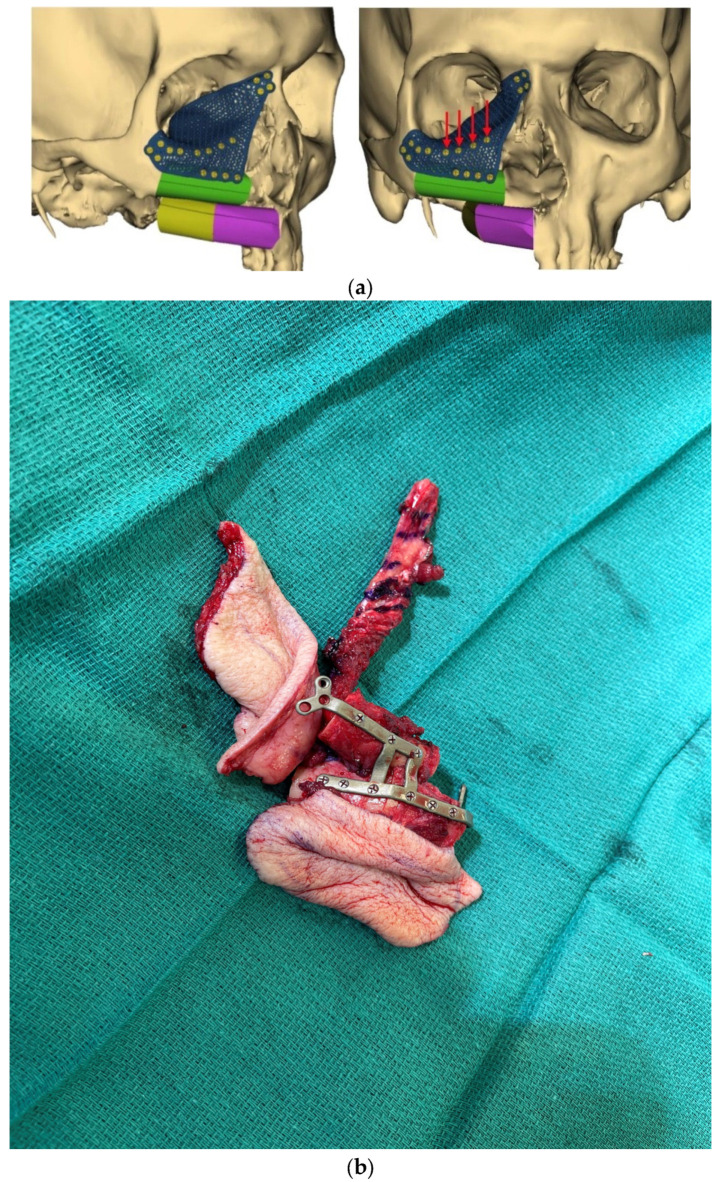
(**a**) Virtual surgical plan with computer-assisted design and computer-assisted modeling (CAD-CAM) for 3D printed midface and orbital floor plate. (**b**) Intra-operative flap fixation to 3D-printed plate with two skin island design.

## Data Availability

No data sharing is available for this manuscript as no new data was gathered in the production of this work.

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
