# Peer review of "Innovative Strategies in Microvascular Head and Neck Reconstruction"

_medicina, 2023, doi:10.3390/medicina59071194_

Round 1

Reviewer 1 Report

I have recently reviewed your paper titled “Innovative Strategies in Microvascular Head and Neck Reconstruction” and would like to provide feedback from a reviewer standpoint.
Firstly, I noticed that the overall organization and layout of the paper could benefit from improvement. Consider arranging the points by operative procedure or flap rather than anatomic region to enhance ease of reading and clarity. This approach would provide a logical flow of information, allowing readers to better understand the context and applicability of the discussed techniques.
Additionally, since this is a review and not a case report the introduction provides a brief overview, but would greatly benefit from expansion. Focus on precise indications for microvascular reconstruction in comparison to pedicled flaps, pros and cons, and ensure that the introduction incorporates up-to-date literature. Complement the existing literature with sources from 2020-2023 to ensure the inclusion of recent advancements in the field. This would improve the manuscript.
Overall, it is important to address limitations in the use of CAD/CAM planned fibula for mandibular replacement in your paper. Specifically, mention cases where the planning process fails to accurately match the intraoperative findings, highlighting the challenges and potential discrepancies that may arise. Additionally, consider discussing the implications of perforator situations and cutting techniques associated with CAD/CAM planned fibula procedures
(See Knitschke et al. https://www.frontiersin.org/articles/10.3389/fonc.2021.821851/full)
This would provide valuable insights into the potential limitations and considerations that surgeons should be aware of when utilizing this technique.
Roy et al. conducted a comprehensive review on gracilis face reinnervation, but your discussion on this operative procedure appears to be brief and resembles a case report. I kindly request you to provide additional details and elaboration on the procedure to enhance the thoroughness and depth of your presentation since this technique is not exactly new.
In line 65, you mentioned that "For the majority of patients who undergo a free fibula flap, radiation is often critical for control of microscopic disease which precludes patients from receiving dental implants.18 However, with the introduction of the next generation of VSP, the potential for dental rehabilitation and improvement of patients' quality of life is achievable. 19-21." It would be beneficial to clarify the relationship between CAD/CAM Fibula and radiotherapy in this context. Specifically, explain how the utilization of CAD/CAM Fibula in conjunction with radiotherapy can potentially overcome the limitations that prevent patients from receiving dental implants, thereby enabling dental rehabilitation and enhancing the quality of life for these individuals.
You mentioned that "Finally, the field of lymphedema supermicrosurgery has also grown tremendously in recent years. Patients previously destined to suffer from the sequelae of lymphedema can now be treated reliably with either a lymphovenous bypass or a vascularized lymph node transfer." It would be
valuable to explore the impact of these advancements in reconstructive surgery specifically in the context of head and neck procedures. While the role of lymphedema supermicrosurgery is well-established for extremities, it is essential to examine its potential relevance in functional reconstruction after tumor resection in the head and neck region and not for breast reconstruction. I encourage you to expand the "future directions" section of your paper, adopting a comprehensive review-style approach. Provide more detailed insights, incorporate relevant literature, and strive for diversity in sources to enrich the discussion and provide a broader perspective on the topic. This will contribute to a more comprehensive understanding of the potential applications of microvascular reconstruction in head neck surgery.

Author Response

I have recently reviewed your paper titled “Innovative Strategies in Microvascular Head and Neck Reconstruction” and would like to provide feedback from a reviewer standpoint.
Firstly, I noticed that the overall organization and layout of the paper could benefit from improvement. Consider arranging the points by operative procedure or flap rather than anatomic region to enhance ease of reading and clarity. This approach would provide a logical flow of information, allowing readers to better understand the context and applicability of the discussed techniques.

We thank the Reviewer for the recommendations and have reorganized the manuscript based on operative procedure which we agree does improve the flow and clarity of the study.

Additionally, since this is a review and not a case report the introduction provides a brief overview, but would greatly benefit from expansion. Focus on precise indications for microvascular reconstruction in comparison to pedicled flaps, pros and cons, and ensure that the introduction incorporates up-to-date literature. Complement the existing literature with sources from 2020-2023 to ensure the inclusion of recent advancements in the field. This would improve the manuscript.

We thank the Reviewer for these important points also. We apologize that we believe we were confined based on the word limit which prevented us from expanding too much on the recommendations. However, for our article, we were granted an increased word limit so are able to expand based on the recommendations. A comparison to pedicle flaps however is beyond the scope of the overview and would represent an entire manuscript with the body of literature on this topic so we opted to focus on the most recent advances in microvascular reconstruction. While not meant to be self-promoting, we did also want to highlight our experience and wanted to focus on some humble advancements that our institution has made in this field. Thank you again for the recommendations.

Overall, it is important to address limitations in the use of CAD/CAM planned fibula for mandibular replacement in your paper. Specifically, mention cases where the planning process fails to accurately match the intraoperative findings, highlighting the challenges and potential discrepancies that may arise. Additionally, consider discussing the implications of perforator situations and cutting techniques associated with CAD/CAM planned fibula procedures
(See Knitschke et al. https://www.frontiersin.org/articles/10.3389/fonc.2021.821851/full)

These are important recommendations which we agree would provide valuable insights into the limitations and considerations that surgeons should be aware of when utilizing this technique. We agree with the Reviewer that this is an important point to include in the manuscript and have expanded on the limitations of CAD-CAM. We agree that there are also some limitations in the imaging in predicting the location of the perforators aside from the limitations of the medical modeling itself. However, the aim of the recommended paper was not necessarily relevant to the technology per se. They did not demonstrate any correlation with the use of CTA predicting perforator locations with flap failure which in our opinion should not have been suspected. Success rates of a free fibula flap are multifactorial, and one would not expect that a CTA would help predict flap survival. However, the same group also published on increased non-union rates with the use of milled titanium plates which we believe is relevant and an important consideration for VSP and medical modeling.

Knitschke M, Sonnabend S, Roller FC, Pons-Kühnemann J, Schmermund D, Attia S, Streckbein P, Howaldt HP, Böttger S. Osseous Union after Mandible Reconstruction with Fibula Free Flap Using Manually Bent Plates vs. Patient-Specific Implants: A Retrospective Analysis of 89 Patients.

Curr Oncol. 2022 May 6;29(5):3375-3392. doi: 10.3390/curroncol29050274.

Roy et al. conducted a comprehensive review on gracilis face reinnervation, but your discussion on this operative procedure appears to be brief and resembles a case report. I kindly request you to provide additional details and elaboration on the procedure to enhance the thoroughness and depth of your presentation since this technique is not exactly new.

We thank the Reviewer for bringing this recent publication to our attention. While this is a thorough review of the gracilis flap for facial reanimation, this is considerably different than our technique. The gracilis flap is not the novelty, but the novelty is specifically for reconstruction following a radical parotidectomy where reconstruction requires not simply facial reanimation but additional soft tissue. A double free flap approach for dynamic facial reanimation to our knowledge has not been thoroughly described or explored, in particular in an elderly population that will require adjuvant radiation. We have provided additional details on this topic.

In line 65, you mentioned that "For the majority of patients who undergo a free fibula flap, radiation is often critical for control of microscopic disease which precludes patients from receiving dental implants.18 However, with the introduction of the next generation of VSP, the potential for dental rehabilitation and improvement of patients' quality of life is achievable. 19-21." It would be beneficial to clarify the relationship between CAD/CAM Fibula and radiotherapy in this context. Specifically, explain how the utilization of CAD/CAM Fibula in conjunction with radiotherapy can potentially overcome the limitations that prevent patients from receiving dental implants, thereby enabling dental rehabilitation and enhancing the quality of life for these individuals.

This is a critical point that the Reviewer astutely points out. We have provided additional insights into this issue where radiation impacts the success of dental implants and the use of CAD-CAM and the potential for achieving dental rehabilitation.

You mentioned that "Finally, the field of lymphedema supermicrosurgery has also grown tremendously in recent years. Patients previously destined to suffer from the sequelae of lymphedema can now be treated reliably with either a lymphovenous bypass or a vascularized lymph node transfer." It would be
valuable to explore the impact of these advancements in reconstructive surgery specifically in the context of head and neck procedures. While the role of lymphedema supermicrosurgery is well-established for extremities, it is essential to examine its potential relevance in functional reconstruction after tumor resection in the head and neck region and not for breast reconstruction.

We again limited our original manuscript based on the word limit, but with the increased allowance, we have added more details on lymphedema surgery in head and neck reconstruction.

I encourage you to expand the "future directions" section of your paper, adopting a comprehensive review-style approach. Provide more detailed insights, incorporate relevant literature, and strive for diversity in sources to enrich the discussion and provide a broader perspective on the topic. This will contribute to a more comprehensive understanding of the potential applications of microvascular reconstruction in head neck surgery.

We thank the Reviewer for the insights and have expanded this section. We also included a broader collection of references in the Introduction to emphasize the contributions made to the field from colleagues around the world as well as in the Future Directions section.

Reviewer 2 Report

hello

thank you for an interesting paper, review 

the abstract is missing key words

citations should be written and cited in the correct mdpi style

very good and illustrative figures

please add limitations in planning with cad-cam

does chimeric flaps usage is more precise when surgery is planned on 3d / cad-cam models?

should major radiation fields in the neck and face be still considered as limitations for microsurgery?

please add how microbiology scrubs before surgery can influence later surgical outcomes

did the anticoagulant protocol over the years had changed significantly to cover the advantages in microsurgery?

after chapter 6 I would add some top key factors that might influence on readers knowledge when using free flaps

thank you 

Author Response

thank you for an interesting paper, review 

the abstract is missing keywords

Thank you for this point. We apologize for this oversight and have added key words to the end of the abstract

citations should be written and cited in the correct mdpi style

We apologize for this oversigh as well and have revised the references into the proper style for the journal.

very good and illustrative figures

Thank you for this kind review.

 please add limitations in planning with cad-cam

We agree there are significant limitations to the use of CAD-CAM technology which we have expanded. We apologize we had limited our discussion with the believe that the word limit was 3500 words, but have been instructed that the limit is 4000.

does chimeric flaps usage is more precise when surgery is planned on 3d / cad-cam models?

We thank the Reviewer for this important question and have attempted to address this issue in the revised manuscript. For most chimeric flaps, there is a soft tissue component that is difficult to plan with the current technology. Most frequently, the fibula flap is used for bony reconstruction, incorporating the use of VSP and 3D modeling. While a chimeric flap can clearly be harvested, the soft tissue is more difficult to plan precisely.

should major radiation fields in the neck and face be still considered as limitations for microsurgery?

This is a very important questions with the changing field of radiation therapy, particularly with the increasing utilization of re-irradiation. While we have demonstrated radiation does not increase the rate of flap losses, the radiation field must be a consideration for all reconstructive microsurgeons. Often the radiated tissue needs to be replaced, the impact on availability of recipient vessels, and the risks of osteoradionecrosis continue to be challenges faced by microsurgeons.

please add how microbiology scrubs before surgery can influence later surgical outcomes

We apologize to the Reviewer, we are not familiar with the impact of “microbiology scrubs” if the Reviewer can provide some more guidance, we would greatly appreciate it.

did the anticoagulant protocol over the years had changed significantly to cover the advantages in microsurgery?

This is a very important point and varies from institution to institution. At the authors’ institution the use of anticoagulation has remained constant for nearly 2 decades. In general, no additional anticoagulation is prescribed except for standard DVT prophylaxis.

after chapter 6 I would add some top key factors that might influence on readers knowledge when using free flaps

We thank the Reviewer for this recommendation and have added some Key points to consider. Thank you.

thank you 

Round 2

Reviewer 1 Report

Thank you for revising your manuscript.